# Deep Transformers with Latent Depth

Xian Li[1], Asa Cooper Stickland[2], Yuqing Tang[1], and Xiang Kong[1]

[1]Facebook AI
{xianl, yuqtang, xiangk}@fb.com
[2]University of Edinburgh
{a.cooper.stickland}@ed.ac.uk

## Abstract

The Transformer model has achieved state-of-the-art performance in many sequence modeling tasks. However, how to leverage model capacity with large or variable depths is still an open challenge. We present a probabilistic framework to automatically learn which layer(s) to use by learning the posterior distributions of layer selection. As an extension of this framework, we propose a novel method to train one shared Transformer network for multilingual machine translation with different layer selection posteriors for each language pair. The proposed method alleviates the vanishing gradient issue and enables stable training of deep Transformers (e.g. 100 layers). We evaluate on WMT English-German machine translation and masked language modeling tasks, where our method outperforms existing approaches for training deeper Transformers. Experiments on multilingual machine translation demonstrate that this approach can effectively leverage increased model capacity and bring universal improvement for both many-to-one and one-to-many translation with diverse language pairs.

## 1 Introduction

The Transformer model has achieved the state-of-the-art performance on various natural language preprocessing (NLP) tasks, originally in neural machine translation [30], and recently in massive multilingual machine translation [3, 37], crosslingual pretraining [8, 17], and many other tasks. There has been a growing interest in increasing the model capacity of Transformers, which demonstrates improved performance on various sequence modeling and generation tasks [35, 24, 1].

Training Transformers with increased or variable depth is still an open problem. Depending on the position of layer norm sub-layer, backpropagating gradients through multiple layers may suffer from gradient vanishing [19, 31, 5]. In addition, performance does not always improve by simply stacking up layers [6, 31]. When used for multilingual or multi-task pretraining, such as multilingual machine translation, crosslingual language modeling, etc., the simplicity of using one shared Transformer network for all languages (and tasks) is appealing. However, how to share model capacity among languages (and tasks) so as to facilitate positive transfer while mitigating negative transfer has not been well explored.

In this work, we present a novel approach to train deep Transformers, in which the layers to be used (and shared) and the effective depth are not static, but learnt based on the underlying task. Concretely, we model the decision to use each layer as a latent variable, whose distribution is jointly learnt with the rest of the Transformer parameters. At training time we approximate the discrete choice with a Gumbel-Softmax [14] distribution. The 'soft weights' sampled from this distribution also act as gradient normalization for each layer, and this allows us to train very deep Transformers (up to 100 layers) without using regular layer normalization layers. At inference time, the learnt discrete choice can be used to directly derive a compact model by pruning layers with low probability, but we

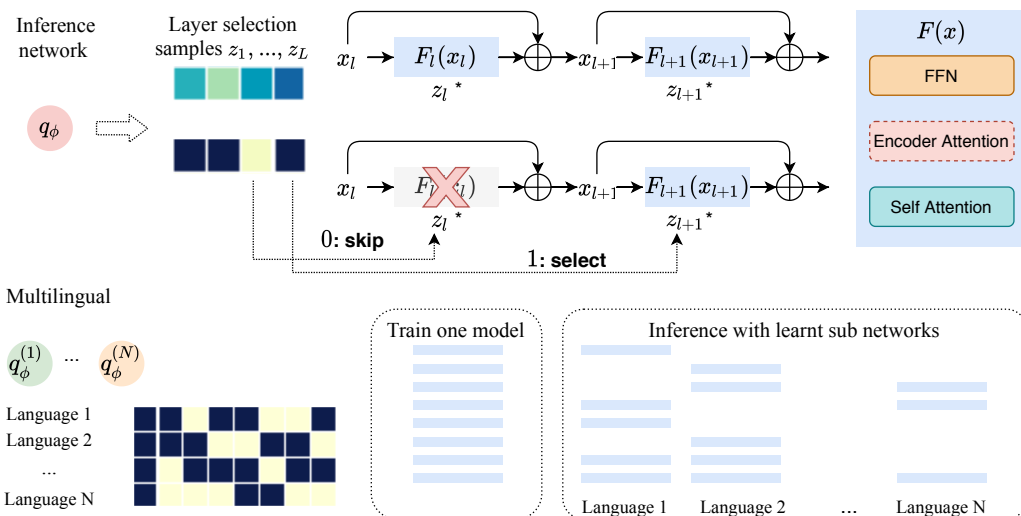

Figure 1: We learn the posterior distribution $q_\phi$ to "select" or "skip" each layer in Transformers. In multilingual setting, each language learns their own "views" of latent layers in a shared Transformer.

have the choice of leaving the learned layer selection probabilities as soft weights. By evaluating on WMT'16 English-German machine translation (MT) and masked language modeling (MLM) tasks (similar to the XLM-R model [8]), we show that we can successfully train deeper Transformer (64-layer encoder/decoder model for MT, and 96-layer encoder for MLM) and outperform existing approaches in terms of quality and training stability.

We show this approach can be extended to learn task-specific sub-networks by learning different layer selection probabilities for each language pair in multilingual machine translation. This result contributes to the growing interest of learning efficient architectures for multi-task and transfer learning in natural language understanding and generation [28, 12, 7].

The main contributions of this paper are as follows. We present a probabilistic framework to learn which layers to select in the Transformer architecture. Based on this framework, we propose a novel method to train one shared Transformer network for multilingual machine translation with different layer selection probabilities for each language pair. The proposed method alleviates the vanishing gradient issue and enables stable training of deep Transformers. We conduct experiments on several tasks to evaluate the proposed approach: WMT'16 English-German machine translation, masked language modeling, and multilingual many-to-one as well as one-to-many machine translation with diverse languages.

## 2 Method

**Background**   In this section, we briefly describe the standard Transformer layer architecture [30]. For a hidden state $x_l$ of a single token at layer $l$, each Transformer layer is a function $F_l(x_l)$ that transforms its input $x_l$ by sequentially applying several sub-layers. The sub-layer is as follows:

$$x_{l+1} = x_l + \text{SubLayer}_l(\text{Norm}(x_l)), \tag{1}$$

where $\text{SubLayer}_l(\cdot)$ is either a Self Attention module, an Encoder Attention module (for a Transformer decoder in a sequence-to-sequence model), or a feed-forward network (FFN) module, and $\text{Norm}(\cdot)$ is a normalisation layer, usually layer-norm [4]. This is the 'pre-norm' setting which is now widely used [19], as opposed to 'post-norm' in which case $\text{Norm}(\cdot)$ would be applied after the residual connection: $x_{l+1} = \text{Norm}(x_l + \text{SubLayer}_l(x_l))$.

## 2.1 Latent Layer Selection

For each Transformer layer $l$, we treat the selection of all sub-layers in non-residual block $F_l(x)$ as a latent variable $z_l$ from a parameterizable distribution $p(z)$,

$$x_{l+1} = x_l + z_l \times F_l(x_l), \, z_l \sim p(z; l) \tag{2}$$

where the standard Transformer [30] is a special case with $z_l = 1$ for $l = 0, ..., L-1$, where $L$ is the depth of the network, i.e. total number of layers.

For the sequence generation task $p(y \mid x)$ parameterized by a Transformer network with the remaining standard parameters $\Theta$, we assume the following generative process:

$$y \sim p(y \mid x; \theta, z), \; p(y \mid x) = \int_z p(y \mid x; \Theta, z) p(\Theta, z) \, \mathrm{d}\Theta \mathrm{d}z \tag{3}$$

**Parameterization and inference of** $z$**.**   We model $z_l$ as discrete latent variable from a Bernoulli distribution with $z_l \sim \mathcal{B}(\pi; l)$, $\pi \in [0, 1]$ indicating select or skip the non-residual block $F_l(x)$ in layer $l$, and samples from one layer are independent from other layers. This modeling choice allows us to prune layers which reduces inference cost and may regularize training.

Marginalizing over $z$ becomes intractable when $l$ grows large. Therefore, we use variational inference as a more general optimization solution. Specifically, we instead maximize the evidence lower bound (ELBO) of Eq. 3

$$\log p(y \mid x) \geq \mathbb{E}_{q_\phi(z)}[\log p_\theta(y \mid x, z)] - \mathrm{D}_{\mathrm{KL}}(q_\phi(z) \parallel p(z)) \tag{4}$$

We point out that although we could treat the rest of the network parameters $\Theta$ as latent variables too and model the joint distribution of $p(\theta, z)$, which could be optimized using Coupled Variational Bayes (CVB) and optimization embedding as demonstrated in [27] for neural architecture search, in practice we found a simpler optimization procedure (Algorithm 2) to learn both $\theta$ and $z$ jointly from scratch.

We use the Gumbel-Softmax reparameterization [14] to sample from the approximate posterior $q_\phi(z)$ which makes the model end-to-end differentiable while learning (approximately) discrete policies without resorting to policy gradients. To allow both "soft weighting" and "hard selection" of layers, each of which has the appealing property of achieving model pruning while training with larger model capacity, we generate soft samples of $z$ during training and draw hard samples for pruning at inference time if $q_\phi(z)$ becomes (close to) discrete. We directly learn the logits parameter $\alpha_l$ for each layer $l$:

$$z_l^i(\alpha_l) = \frac{\exp((\alpha_l(i) + \epsilon(i))/\tau)}{\sum_{i \in \{0,1\}} \exp((\alpha_l(i) + \epsilon(i))/\tau)} \, , \epsilon \sim \mathcal{G}(0, 1) \tag{5}$$

where $\mathcal{G}(0, 1)$ is the Gumbel distribution, and $\tau$ is a temperature hyperparameter which increases the discreteness of samples when $\tau \to 0$. For $p(z)$ we can use the conjugate prior Beta$(a, b)$ which allows us to express different preferences of $z$, such as $a = b = 1$ for an uniform prior, $a > b$ to bias towards layer selection and $a < b$ to favor skipping layers.

**Gradient scaling.**   Next we analyze the impact of latent layers on gradient backpropagation during training in the pre-norm setting. In Eq. 6, we can see that given the forward pass loss $\mathcal{L}$, the gradient accumulation from higher layers $m_{l<m<L}$ is now weighted by the their corresponding latent samples $z_m$, which acts as gradient scaling. In Section 3 we show that with such gradient normalization we can train deeper Transformers without using layer normalisation.

$$\frac{\partial \mathcal{L}}{\partial x_l} = \frac{\partial \mathcal{L}}{\partial x_L} \times (1 + \sum_{m=l}^{L-1} z_m \frac{\partial F_m(x_m)}{\partial x_l}) \tag{6}$$

## 2.2 Multilingual Latent Layers

It is sometimes convenient to share a Transformer network across multiple languages, enabling crosslingual transfer, with recent success in multilingual machine translation and multilingual pre-training (e.g. multilingual BERT and BART) [3, 8, 20, 17]. Current approaches share a vanilla (usually 12-layer) Transformer across all languages.

To explore the potential of latent layers for a multilingual Transformer, we let each language learn its own layer utilization given a single Transformer network $\Theta$ shared among $N$ languages by learning its own posterior inference network $q_\phi^{(n)}$ of $\{\alpha_l\}$. We acknowledge that an alternative is to learn a shared inference network $q_\phi(n)$ which takes language $n$ as input. The latter may enable learning commonalities across languages but at the cost of extra parameters, including a non-trivial $N \times d$ parameters for language embeddings. Therefore, we chose the former approach and leave the latter (and the comparison) for future work. With this modeling choice, we can still encourage layer-sharing across languages by using the aggregated posterior across languages $\tilde{q}(z)$ as the prior in the $D_{KL}$ term:

$$\mathrm{D_{KL}}(q_\phi(z) \parallel \tilde{q}(z)) = \mathbb{E}_{q_\phi(z)}[\log \frac{q_\phi(z)}{\tilde{q}(z)}] \, , \, \tilde{q}(z) = \frac{1}{N} \sum_{n=1}^{N} q_\phi(z \mid x^{(n)}, y^{(n)}, \hat{\theta}) \tag{7}$$

**Latent Layers with Targeted Depth** To deploy Transformers in the real world, we would like to have lower computational cost at inference time. Within a Transformer layer, some computation is parallel, such as multi-head attention, but the time and space complexity at inference time grows linearly with the number of layers. Therefore, pruning layers at test time directly reduces inference cost. Our approach can be extended to perform model pruning, encouraging the model to achieve a target depth $K$ by adding an extra loss $\mathcal{L}_K = \| \sum_{l=0}^{L-1} u_l - K \|_2$ where $u_l$ refers to the "utilization" of layer $l$. $u_l$ can be approximated by samples of the latent variables $z_l$ and for the multilingual case $u_l = \sum_{n=1}^{N} z_l^{(n)}/N$.

The general loss for training a Transformer with latent depth $K$ is

$$\mathcal{L}_{LL} = \underbrace{\mathbb{E}_{q_\phi(z)}[- \log p_\theta(y \mid x, z)] + \beta \, \mathrm{D_{KL}}(q_\phi(z) \parallel p(z))}_{\mathcal{L}_{\mathrm{ELBO}}} + \lambda \mathcal{L}_K \tag{8}$$

To learn $\Theta$ and $q_\phi$ jointly from scratch, we use an two-level optimization procedure described in Algorithm 2. This training strategy is inspired by the Generalized Inner Loop Meta-learning [10]. We provide a more detailed explanation of this training procedure in Appendix B.1.

## 3 Experimental Settings

We first evaluate on the standard WMT English-German translation task and a masked language modeling task to demonstrate the effectiveness of the proposed approach at enabling training deeper Transformers and whether this increased depth improves model performance. We then evaluate multilingual latent layers (see section 2.2) on multilingual machine translation.

**Bilingual Machine Translation.** We use the same preprocessed WMT'16 English-German sentence pairs as is used in [30, 31]. To make comparison more clear and fair, we evaluate on the last model checkpoint instead of ensembles

---

**Algorithm 1** Training with Latent Layers

1: Initialize $\Theta$, $q_\phi$.
2: **for** t=1, ..., T **do**
3:      **for** i=1, ..., I **do**
4:          Sample a mini-batch $(x, y) \sim D$ .
5:          Sample $z_{l=0,...,L-1}$ with Eq. 5
6:          Compute     $\hat{\mathcal{L}}_{LL}((x,y); \Theta_{i-1}, q_\phi^{t-1})$
     with Eq. 8
7:          Update $\Theta_i = \Theta_{i-1} - \eta \nabla_{\Theta_{i-1}} \hat{\mathcal{L}}_{LL}$
8:      Update $q_\phi^t = q_\phi^{t-1} - \eta \nabla_{q_\phi^{t-1}} \hat{\mathcal{L}}_{LL}$

---

from averaging the last 5 checkpoints. We use beam size 5 and length penalty 1.0 in decoding and report corpus-level BLEU with sacreBLEU [22].

**Crosslingual Masked Language Modelling.** We test our method on a scaled-down version of XLM-R [8], intending to show the promise of our method, but not obtain state-of-the-art results on downstream tasks. In particular we use as training data the Wikipedia text of the 25 languages used in the mBART [17] model, and evaluate using perplexity on a held out dataset consisting of 5000 sentences in each language (sampled randomly from each Wikipedia text).

**Multilingual Machine Translation.** We evaluate the proposed approach on multilingual machine translation using the 58-language TED corpus [23]. To study its performance independent of task

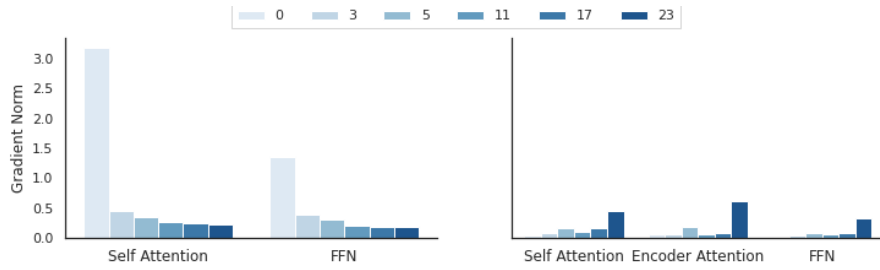

(a) Gradient norms of encoder and decoder in standard Transformer.

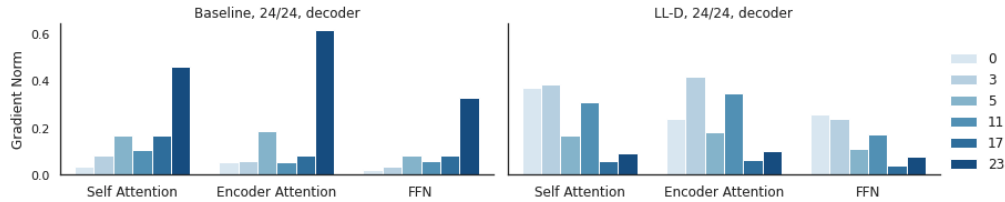

(b) Improvement of decoder's gradient norm using latent layers.

Figure 2: Comparing gradient norms of baseline (a) and using latent layers (b).

similarity and difficulty, we evaluate on both *related* (four low resource languages and four high resource languages from the same language family) and *diverse* (four low resource languages and four high resource ones without shared linguistic properties) settings as is used in [32]. Dataset descriptions and statistics are summarized in the Appendix C.1. For each set of languages, we evaluate both many-to-one (M2O), i.e. translating all languages to English, and one-to-many (O2M), translating English to each of the target languages, which is a more difficult task given the diversity of target-side languages.

**Baselines.** We compare to the standard Transformer with static depth on machine translation task and "wide" model, e.g. Transformer-big architecture in [30] which increases the hidden (and FFN) dimension and has been a common approach to leverage large model capacity without encountering the optimization challenges of training a deeper model.

We also compare to recent approaches to training deeper Transformers:

- Random Layer drop. For deeper models where the static depth baselines diverged, we apply the random LayerDrop described in [9] which trains a shallower model by skipping layers.
- Dynamic linear combination of layers (DLCL). This is a recently proposed approach to address vanishing gradient by applying dense connections between layer which was demonstrated effective for machine translation[31].
- ReZero[5]. This is similar to our method in that both methods learn to weigh each layer. The key difference is that ReZero learns (unconstrained) weighting parameters. In our experiments, we found ReZero suffers from gradient exploding and training loss diverged.

# 4   Results

## 4.1   Addressing vanishing gradient

First, we empirically show that with static depth, gradient vanishing happens at bottom layers of decoder Figure 2a. The effect of training with latent layers using the proposed approach is illustrated in Figure 2b, which shows that gradient norms for bottom layers in the decoder are increased.

Next, we compared the learning curves when training deeper models. As is shown in Figure

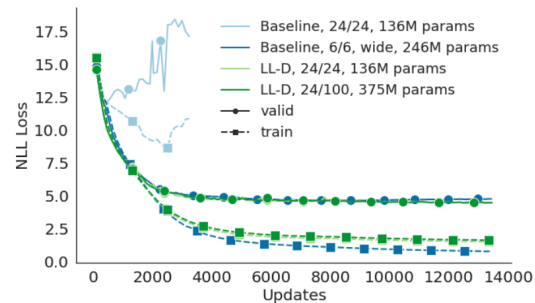

Figure 3: Comparing learning curves, training and validation per-token negative loglikelihood (NLL) loss, of baseline models (static depth) and the proposed method when training deeper model (decoder).

3 (evaluated on multilingual translation task O2M-Diverse dataset), the baseline model with static depth diverged for a 24-layer decoder, while using the latent layers ((LL-D) approach we could train both 24-layer and 100-layer decoder successfully. We further compared the 100-layer model with a wider model (Transformer-big), and found that besides stable training, deep latent layer models are less prone to overfitting (i.e. they achieve lower validation loss, with a smaller gap between train and validation losses) despite having more parameters.

## 4.2   En-De Machine Translation

In Table 1 we evaluate on training deeper Transformers and examine the impact of latent layers in decoder (LL-D) and both encoder and decoder (LL-Both) respectively. Compared to existing methods for training deeper Transformers such as using dense residual connections (DLCL), our approach can leverage larger model capacity from increased depth and achieved improved generalization.

| Model | Params | $\text{NLL}_{\text{valid}} \downarrow$ | $\text{BLEU}_{valid} \uparrow$ | $\text{BLEU}_{test} \uparrow$ |
|---|---|---|---|---|
| Transformer-Big | 246M | 2.081 | 28.7 | 28.1 |
| DLCL, 36/36 | 224M | 2.128 | 28.5 | 27.7 |
| DLCL, 48/48 | 224M | 2.090 | **28.8** | 28.1 |
| LL-D, 12/24 | 135M | 2.179 | 28.1±0.08 | 27.2±0.04 |
| LL-D,12/48 | 211M | 2.128 | 28.1±0.00 | 27.3±0.04 |
| LL-Both, 36/36 | 224M | 2.147 | 28.4±0.07 | 28.1±0.07 |
| LL-Both, 48/48 | 287M | 2.078 | 28.7±0.10 | **28.7±0.09** |
| LL-Both, 64/64 | 371M | **2.069** | 28.5±0.07 | 28.4±0.08 |

Table 1: Performance on WMT'16 En-De. For BLEU scores evaluation, we provide standard errors from 5 runs with different seeds.

## 4.3   Masked Language Modeling

Latent layers (LL) is also shown to be effective for training deeper encoder without divergence (see Table 2). For 24 and 48 layer encoders, we observed stable training with 2x learning rate and achieved better performance for 24 layers. However the result of scaling up to 96 layers was slightly worse performance than a vanilla 48 layer model. This shows the promise of the method for stabilising training at increased depth, however we did not attempt to scale up our data to match our larger model capacity.

| Model | Params | Perplexity $\downarrow$ |
|---|---|---|
| Static depth 24 | 202M | 2.91 |
| LL, 24 | 202M | 2.82 |
| Static 48 | 372M | 2.60 |
| LL, 48 | 372M | 2.71 |
| Static 96 | 712M | Diverged |
| + layer-drop | 712M | Diverged |
| LL, 96 | 712M | 2.66 |

Table 2: Perplexity on held-out data for crosslingual masked language modeling.

## 4.4   Multilingual Translation

We separately test the impact of applying latent layers in the decoder (LL-D), encoder (LL-E) and both (LL-Both).

| Model | Params | Avg. | aze | bel | ces | glg | por | rus | slk | tur |
|---|---|---|---|---|---|---|---|---|---|---|
| 6/6 | 63.6M | 19.65 | 5.4 | 9.1 | 21.9 | 22.4 | 38.6 | 19.4 | 24.6 | 15.8 |
| 6/6, wide | 190M | 20.33 | **5.7** | 9.7 | 22.4 | 23.1 | 40.3 | 20.6 | 24.1 | 16.8 |
| 12/12 | 95.1M | 20.48 | 5.6 | 10.3 | 23.1 | 22.8 | 39.7 | 20.1 | **25.1** | 17.1 |
| 12/24 | 133M | NA | - | - | - | - | - | - | - | - |
| 24/24 | 158M | NA | - | - | - | - | - | - | - | - |
| +layer drop | 158M | 11.16 | 3.3 | 7.5 | 11.6 | 14.4 | 23.4 | 10.4 | 12.9 | 5.8 |
| LL-D, 12/24 | 133M | 20.83 | 5 | 10.2 | 23.4 | **24.3** | 40.3 | **21** | 24.8 | **17.6** |
| LL-D, 24/24 | 158M | **20.84** | 5.3 | **10.6** | **23.4** | 23.7 | **40.7** | 20.9 | 24.8 | 17.5 |

Table 3: BLEU scores for one-to-many multilingual translation on related languages. "NA" means training diverged.

**Latent layers in decoder.**   To evaluate the impact of increased decoder depth, we tested on one-to-many (O2M) multilingual translation. In Table 3 we show performance on the "Related" languages setting. Baseline models began to diverge when decoder depth increases to $L = 24$, and applying random LayerDrop did not help. Latent layers allows us to train the same depth successfully, and we observe improvement in translation quality for both language pairs as well as overall quality shown by the average BLEU score. In Table 4, we evaluate the impact of deeper decoder with latent layers

in the O2M-Diverse setting. This is a more challenging task than O2M-Related since decoder needs to handle more diversified syntax and input tokens.

| Model | Avg. | bos | mar | hin | mkd | ell | bul | fra | kor |
|---|---|---|---|---|---|---|---|---|---|
| 6/6 | 22.12 | 12.6 | 11.1 | 14.6 | 22.7 | 29.8 | 31.8 | 37.3 | 17.1 |
| 6/6, wide | 23.51 | 12.7 | 11.3 | 13.9 | 23.8 | 32.5 | 34.8 | 40.6 | 18.5 |
| 12/12 | 23.34 | 13.1 | 11.1 | 13.6 | 22.5 | 32.7 | 34.7 | 40.4 | 18.6 |
| 12/24 | NA | - | - | - | - | - | - | - | - |
| 24/24 | NA | - | - | - | - | - | - | - | - |
| +layer drop | 22.06 | 13.0 | 10.0 | 12.2 | 21.5 | 30.7 | 33.0 | 38.5 | 17.6 |
| LL-D, 12/24 | 23.70 | 13.4 | 10.7 | 14.1 | 22.8 | 33.1 | 35.1 | 41.1 | 19.3 |
| LL-D, 12/100 | 24.16 | 13.5 | 10.6 | 13.8 | 24.1 | 32.7 | **38.2** | 41.3 | 19.1 |
| LL-D, 24/24 | **24.46** | **15.5** | **11.4** | **14.6** | **24.4** | **33.5** | 35.5 | **41.5** | **19.3** |

Table 4: BLEU scores for one-to-many multilingual translation on diverse languages.

**Latent layers in encoder, decoder, and both.** We use the many-to-one multilingual translation task to verify the pattern observed above, and test the effect of increased depth in encoder. Results are summarized in Table 5. Similar to O2M, standard Transformer begins to diverge when decoder depth increase over 24 while applying latent layers enable successful training and yields improved translation quality.

| Model | Avg. | bos | mar | hin | mkd | ell | bul | fra | kor |
|---|---|---|---|---|---|---|---|---|---|
| 6/6 | 25.95 | 20.7 | 8.6 | 19.2 | 30.0 | 36.3 | 36.9 | 38.4 | 17.5 |
| 12/12 | 27.73 | 22.5 | 9.4 | 20.1 | 31.6 | 38 | 39.6 | 40.8 | 19.9 |
| 24/12 | 27.86 | 23.7 | 9.7 | 21.6 | 31.2 | 37.6 | 39.3 | 40.0 | 19.8 |
| 24/24 | NA | - | - | - | - | - | - | - | - |
| +layer drop | 26.7 | 21.3 | 9 | 19.2 | 29.2 | 37.5 | 38.8 | 39.9 | 18.7 |
| LL-E, 36/12 | 27.98 | **24.2** | 10.2 | 21.9 | 32 | 37.3 | 38.8 | 39.3 | 20.1 |
| LL-D, 12/24 | 27.63 | 22.4 | 9.3 | 20.2 | 30.8 | 38.2 | 39.7 | 40.5 | 19.9 |
| LL-D, 12/36 | 27.89 | 22.3 | 9.5 | 21.1 | 30.7 | 38.2 | 40.2 | **41.2** | 19.9 |
| LL-D, 24/24 | 28.43 | 23.6 | 10.0 | 21.9 | 31.7 | **38.4** | 40.3 | **41.2** | 20.4 |
| LL-Both, 24/24 | **28.56** | 23.5 | 10.3 | **22.3** | **32.8** | 38.3 | 40 | 40.8 | **20.5** |

Table 5: BLEU scores of models with increased depth in the encoder and decoder for many-to-one on diverse languages.

By applying latent layers to encoder only (LL-E) we found increased depth (36/12) improves low resource languages (e.g. bos and hin) over the baseline (12/12). However, deeper decoder (12/36) or even allocation of depth (24/24) brings consistent gains as is shown in Fig 4.

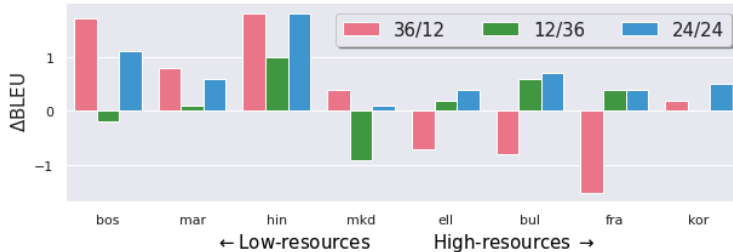

Figure 4: Quality improvement (over static depths 12/12) by allocating increased capacity to all-encoder (36/12), all-decoder (12/36), and even allocation (24/24).

## 5 Analysis

In this section, we analyze the effect of several modeling choices and understand their contribution to the results.

**Effect of Priors** In Figure 5 we illustrate the difference between using aggregated posterior $\tilde{q}(z)$ versus a uniform prior $\text{Beta}(1, 1)$ in computing the $\text{D}_{\text{KL}}$ loss.

Compared to the uniform prior, using the aggregated posterior as prior discretizes layer utilization, that is, the model is incentivised to make layer selections consistent across all languages, i.e. facilitating parameter sharing. Interestingly, the learnt "sharing" pattern by using $\tilde{q}(z)$ as prior is consistent with

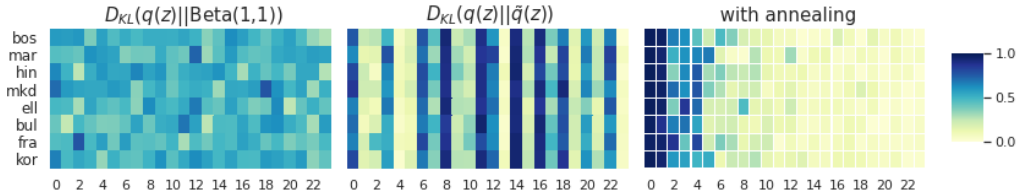

Figure 5: Layer selection samples $z_l$ at epoch 1 from different priors used for $D_{KL}$.

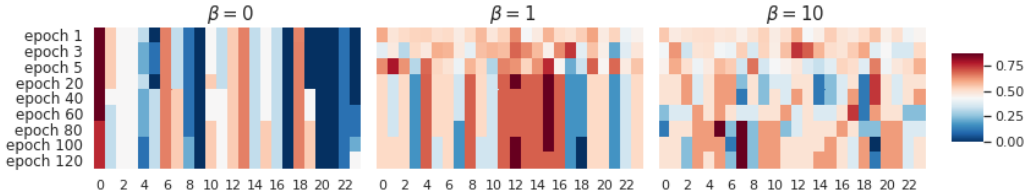

Figure 6: Visualization of layer utilization $u_l$ during training using the M2O Diverse dataset.

heuristics such as dropping every other layer for pruning which was empirically found effective [9]. However, training with such a prior in the beginning can lead to "posterior collapse", which is a well-known challenge found in training variational autoencoders. After applying "KL annealing" (annealing the $D_{KL}$ coefficient $\beta$), we can see that layer selection samples are more continuous with a curriculum to use the bottom layers first.

**Effect of $\beta$.** In order to understand how the $D_{KL}$ loss term affects layer selection policies and samples *throughout* training, we vary the $D_{KL}$ coefficient $\beta \in \{0, 1, 10\}$. First, we examine layer utilization $u_l$, e.g. whether "hot layers" ($u_l \rightarrow 1$) and "cold layers" ($u_l \rightarrow 0$) change over time. As is shown in Figure 6, without the $D_{KL}$ term, layer utilization stays constant for most of the layers, especially several top layers whose parameters were rarely updated. By increasing the contribution from the $D_{KL}$ to the total

|  | $\mathbb{E}_L$ | Avg. valid BLEU |
|---|---|---|
| $\beta = 0$ | 10.25 | 28.50 |
| $\beta = 1$ | 11.25 | 28.53 |
| $\beta = 10$ | 12.125 | 28.23 |

Table 6: Impact of the KL coefficient $\beta$ on network effective depth ($\mathbb{E}_L$) and translation quality, evaluated on M2O-Diverse.

loss, layer selections are more evenly spread out across languages, i.e. $u_l$ becomes more uniform. This is also reflected in Table 6 where the "effective depth" $\mathbb{E}_L$ increases with $\beta$.

## 5.1 Ablation Studies

In this section, we provide ablation experiments to understand how different loss terms contribute to the results. Table 7 compares the effect on translation quality from different loss terms in Eq. 8. We can see that optimizing the $\mathcal{L}_{ELBO}$ loss brings the most quality gains, and $\mathcal{L}_K$ loss adds additional improvement by acting as a regularization.

| Model | Avg. | bos | mar | hin | mkd | ell | bul | fra | kor |
|---|---|---|---|---|---|---|---|---|---|
| LL-D, 24/24 | **24.46** | **15.5** | **11.4** | 14.6 | 24.4 | **33.5** | **35.5** | 41.5 | **19.3** |
| - $\mathcal{L}_K$ | 24.28 | 14.5 | 10.8 | 14.3 | **25** | 33.4 | 35.4 | **41.6** | 19.2 |
| - $D_{KL}$ | 23.89 | 13.7 | 11.0 | **14.7** | 24.2 | 32.9 | 35.0 | 40.9 | 18.9 |
| - both | 23.75 | 13.4 | 10.9 | 14.2 | 23.6 | 33.2 | 35.2 | 40.7 | 18.8 |

Table 7: Effects from different terms in $\mathcal{L}_{LL}$ evaluated on the O2M-Diverse dataset.

## 5.2 Latent depth vs. static depth

We compare a deeper model with latent effective depth $\mathbb{E}[L]$ to models with the same depth trained from scratch.

As is observed in both bilingual (Table 8) and multilingual (Table 9) machine translation tasks, training a deeper model with latent depth outperforms standard Transformer with the same number of effective layers but trained with static depth.

| Model | BLEU$_{valid}$ ↑ | BLEU$_{test}$ ↑ |
|---|---|---|
| Latent depth, $L = 24$, $\mathbb{E}[L] = 12$ | **28.6**±**0.07** | **27.88**±**0.04** |
| Static depth, $L = 12$ | 27.2 | 26.5 |

Table 8: Comparing a 24 latent layers model with effective depth $\mathbb{E}[L] = 12$ with a 12-layer static depth model trained from scratch, evaluated on WMT'16 En-De.

| Model | Avg. | bos | mar | hin | mkd | ell | bul | fra | kor |
|---|---|---|---|---|---|---|---|---|---|
| Latent depth, $\mathbb{E}[L] = 14.5$ | **28.43** | 23.6 | 10.0 | **21.9** | **31.7** | **38.4** | **40.3** | **41.2** | **20.4** |
| Static depth, $L = 15$ | 27.9 | **23.9** | **10.3** | 21.5 | 31.4 | 37.5 | 38.9 | 39.8 | 19.9 |

Table 9: Comparing a 24 latent layers model with effective depth $\mathbb{E}[L] = 14.5$ with a 15-layer static depth model trained from scratch, evaluated on M2O-Diverse dataset.

# 6 Related Work

The Transformer model [30] has achieved state-of-the-art performance on various natural language processing (NLP) tasks. Theoretical results suggest networks often have an expressive power that scales exponentially in depth instead of width [21], and recent work [36, 1, 23, 32] finds that deeper Transformers improve performance on various generation tasks. However, deeper Transformer models also face the gradient vanishing/exploding problem leading to unstable training [6, 31]. In order to mitigate this issue, Huang et al. (2016) [13] drop a subset of layers during the training, and bypass them with the identity function. Zhang et al. (2019) [38] propose an initialization method to scale gradients at the beginning of training to prevent exploding or vanishing gradient. Bachlechner et al. (2020) [5] initialize an arbitrary layer as the identity map, using a single additional learned parameter per layer to dynamically facilitates well-behaved gradients and arbitrarily deep signal propagation. Fan et al. (2019) [9] introduce a form of structured dropout, *LayerDrop*, which has a regularization effect during training and allows for efficient pruning at inference time. Concurrent work which shown improvement on NMT task by increasing model depth includes Zhang et al. (2020) [37] and Wei et al. (2020) [33].

Exploring dynamic model architecture beyond hard parameter sharing has received growing interest. In multi-task learning, Multi-Task Attention Network (MTAN) [16], routing network [25] and branched network [29] enables soft parameter sharing by learning a dynamic sub-network for a given task. One concurrent work "GShard" [15] also demonstrate deeper model with conditional computation brings consistent quality improvement for multilingual translation. More work on learning an adaptive sub-network includes BlockDrop [34] which learns dynamic inference paths per instance, and SpotTune [11] which learns which layers to finetune or freeze to improve transfer learning from a pretrained model.

# 7 Conclusion

We proposed a novel method to enable training deep Transformers, which learns the effective network depth, by modelling the choice to use each layer as a latent variable. Experiments on machine translation and masked language modeling demonstrate that this approach is effective in leveraging increased model capacity and achieves improved quality. We also presented a variant of this method in a multilingual setting where each language can learn its own sub-network with controllable parameter sharing. This approach can be extended to use a shared Transformer for multi-task learning in NLP tasks, and offers insight into which layers are important for which tasks.

# Broader Impact

This work proposes a new method to leverage a model with increased depth during training, while learning a compact sub-work with reduced depth which can be used for deployment in real-world applications where Transformers have achieved state-of-the-art quality such as machine translation systems, dialog and assistant applications, etc, as reducing the number of layers especially in

decoder (often autoregressive) can have direct impact on reducing inference-time latency, memory consumption, etc. However scaling up the number of layers adds to energy cost of training, even if we can prune at inference time.

We hope our research on multilingual NLP will contribute to the effort of improving the standard of NLP tools for low-resource languages. However we only test our machine translation systems on to-English or from-English tasks, leaving out translation from non-English languages to other non-English languages entirely.

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
