[Supplementary Material]

# A Gradient analysis

We provide a detailed derivation of Eq. 6. The gradient backpropagated to layer $l$, $\frac{\partial \mathcal{L}}{\partial x_l}$, can be computed by applying the chain rule:

$$\frac{\partial \mathcal{L}}{\partial x_l} = \frac{\partial \mathcal{L}}{\partial x_L} \times \frac{\partial x_L}{\partial x_l} \tag{9}$$

$$\tag{10}$$

To compute $\frac{\partial x_L}{\partial x_l}$, we first apply Eq. 2 recursively to expand $x_L$ as:

$$x_L = x_{L-1} + z_{L-1} \times F_{L-1}(x_{L-1}) \tag{11}$$

$$= x_{L-2} + z_{L-2} \times F_{L-2}(x_{L-2}) + z_{L-1} \times F_{L-1}(x_{L-1}) \tag{12}$$

$$= x_l + \sum_{m=l}^{L-1} z_m F_m(x_m) \tag{13}$$

$$\frac{\partial x_L}{\partial x_l} = 1 + \sum_{m=l}^{L-1} z_m \frac{\partial F_m(x_m)}{\partial x_l} \tag{14}$$

$$\frac{\partial \mathcal{L}}{\partial x_l} = \frac{\partial \mathcal{L}}{\partial x_L} \times (1 + \sum_{m=l}^{L-1} z_m \frac{\partial F_m(x_m)}{\partial x_l}) \tag{15}$$

# B Training Details

## B.1 Training procedure

The proposed training procedure is motivated by the Generalized Inner Loop Meta-learning [10] although we use first-order gradient as approximation. Specifically, we treat $q_\phi$ as "meta parameters" and the rest of the Transformer parameters $\Theta$ as "task-specific" parameters. A key difference is that in our case there is only one task and the support set and target set are from the same distribution. At a high-level, we learn $\Theta$ in an inner-loop while updating $q_\phi$ from the unrolled gradient steps. Such nested optimization is computationally expensive as the graph for multiple steps needs to be stored in memory, and training was found to be unstable due to challenges in backpropagating second-order gradients through multiple steps [2]. We adopt a multi-step loss approximation using first-order gradients only as is shown to be effective in [2]. Specifically, in each outer loop we take the latest parameters of $q_\phi^{t-1}$, and perform $I$ inner loop steps. The gradients from each inner loop loss $\hat{\mathcal{L}}$ are directly backpropagated to $\Theta$, and the last step's gradient are used to update $q_\phi$, which is a special case of multi-step loss annealing where $\omega_{I-1} = 1, \omega_j = 0$ for $j < I - 1$.

---

**Algorithm 2** Training with latent layers in multilingual setting

---

1: **Input:** training examples from $N$ languages $\{D_n\}_{n=1}^N$; total number of training steps $T$; inner loop update frequency $I$
2: Initialize $\Theta$, $q_\phi^0 = \{\alpha_l^0\}$; $t = 0$.
3: **for** t=1, ..., T **do**
4:     **for** i=1, ..., I **do**
5:         **for** n = 1, ..., N **do**
6:             Sample a mini-batch $(x, y) \sim D_n$.
7:             Compute $z_{l=0,...,L-1}$ all at once following Eq. 5 with samples $\epsilon_l \sim \mathcal{G}$
8:             Compute loss $\hat{\mathcal{L}}_{LL}((x, y); \Theta_{i-1}, q_\phi^{t-1})$ with Eq. 8
9:         Update $\Theta_i = \Theta_{i-1} - \eta \nabla_{\Theta_{i-1}} \hat{\mathcal{L}}_{LL}$
10:     Update $q_\phi^t = q_\phi^{t-1} - \eta \nabla_{q_\phi^{t-1}} \hat{\mathcal{L}}_{LL}$

---

## B.2 Training stability.

We examine the stability of our training procedure, e.g. whether training is sensitive to the choice of inner loop frequencies. Figure 7 plots the gradient norms of using $I \in \{1, 2, 5, 10\}$, and the impact on translation performance is summarized in Table 10.

## C  Experiments Implementation Details

### C.1  Dataset description

For WMT'16 English-German experiment, we used the same preprocessed data provided by [31] [1], including the same validation (*neewsteest2013*) and test (*neewsteest2014*) splits. The data volume for train, validation and test splits are 4500966, 3000, 3003 sentence pairs respectively. The data was tokenized and numberized with a joint BPE (byte pair encoding) [26] vocabulary with 32k merge operations.

Figure 7: Comparison of gradient norms using different inner loop iterations $I$ to verify training stability is not sensitive to the choice of $I$.

For multilingual translation experiments, we use the same preprocessed data[2] provided by [32] using the same train, valid, and test split as in [23]. The data volumes for related and diverse language groups are summarized in Table 12.

For crosslingual language modelling we used data from Wikipedia from the 25 languages used in the mBART [17] model, using the same data collection and preprocessing as [8]. We list the languages used and corresponding Wikipedia corpus size in Table 11. A random sample of 5000 sentences from each of the languages was used as held-out data to compare models.

|          | Avg.  | bos  | mar  | hin  | mkd  | ell  | bul  | fra  | kor  |
|----------|-------|------|------|------|------|------|------|------|------|
| $I=1$    | 28.28 | 23.0 | 14.1 | 19.2 | 31.6 | 37.2 | 39.4 | 40.8 | 20.9 |
| $I=2$    | 28.49 | 23.4 | 15.1 | 19   | 32.1 | 37.2 | 39.4 | 40.8 | 20.9 |
| $I=5$    | 28.24 | 23.4 | 14.5 | 18.6 | 32.1 | 37.2 | 39.1 | 40.2 | 20.8 |
| $I=10$   | 28.25 | 23.8 | 14.3 | 19   | 314  | 37   | 39.4 | 40.5 | 20.6 |

Table 10: BLEU scores on validation set to assess the impact of the inner loop frequency $I$ on training stability and model performance, evaluated on the M2O-Diverse dataset.

### C.2  Models and hyperparameters

Both baselines and proposed models are implemented using Transformer models in fairseq [18]. For baseline models, we use the pre-norm setting which provides a stronger baseline since it was shown to more effective for training deeper Transformer models than post-norm[19, 31]. Therefore, the comparison with baseline can focus on evaluate the difference made from using latent layers. We use per-token negative loglikelihood (NLL) loss on the validation set to choose the loss coefficients for $\beta$ and $\lambda$.

**WMT'16 English-German.**  All models were trained for 75 epochs and evaluating on the last checkpoint. For Transformer-big, we use the standard model architecture as is described in [30]: $d = 1024$ for embedding and hidden dimension, and $d = 4096$ for FFN dimension, 6-layer encoder and decoder, 0.3 dropout (0.1 after attention sub-layer and ReLU activation). Model was trained with 8192 token per GPU and 32 GPUs, learning rate 7e-4 and 8000 warm-up updates with Adam optimizer. For deeper models, i.e. both DLCL (baseline) and latent layers (LL, the proposed approach), since the depth is increased we reduce the model width by using $d = 512$ for embedding and hidden dimension, and $d = 1024$ for FFN dimension, and 4 attention heads. Also, we found for deeper models we were able to use almost $2\times$ learning rate (1.5e-3). We use $\beta = 1$ and $\lambda = 0.1$ for latent layers models.

**Crosslingual Masked Language Modelling.**  We use the XLM-R$_{\text{Base}}$ architecture of [8], which has a hidden dimension of 768, but we explore increasing the number of layers, considering 24, 48 and 96 layer

| Code | Language | Sentences (M) |
|---|---|---|
| **En** | English | 41.9 |
| **Ru** | Russian | 12.0 |
| **Vi** | Vietnamese | 3.7 |
| **Ja** | Japanese | 1.7 |
| **De** | German | 16.7 |
| **Ro** | Romanian | 1.8 |
| **Fr** | French | 14.8 |
| **Fi** | Finnish | 2.4 |
| **Ko** | Korean | 2.1 |
| **Es** | Spanish | 10.9 |
| **Zh** | Chinese (Sim) | 5.2 |
| **It** | Italian | 9.7 |
| **Nl** | Dutch | 7.7 |
| **Ar** | Arabic | 3.2 |
| **Tr** | Turkish | 1.8 |
| **Hi** | Hindi | 0.6 |
| **Cs** | Czech | 2.7 |
| **Lt** | Lithuanian | 0.9 |
| **Lv** | Latvian | 0.45 |
| **Kk** | Kazakh | 1.0 |
| **Et** | Estonian | 2.2 |
| **Ne** | Nepali | 0.1 |
| **Si** | Sinhala | 0.1 |
| **Gu** | Gujarati | 0.1 |
| **My** | Burmese | 0.4 |

Table 11: A list of the 25 languages and corresponding Wikipedia corpus size (in millions of sentences) used for crosslingual masked language modelling.

| | **Related** | | | | | | | | **Diverse** | | | | | | | |
|---|---|---|---|---|---|---|---|---|---|---|---|---|---|---|---|---|
| | **aze** | **bel** | **glg** | **slk** | **ces** | **por** | **rus** | **tur** | **bos** | **mar** | **hin** | **mkd** | **ell** | **bul** | **fra** | **kor** |
| **train (K)** | 5.94 | 4.51 | 10 | 61.5 | 103 | 195 | 208 | 182 | 5.64 | 9.84 | 18.79 | 25.33 | 134 | 174 | 192 | 205 |
| **valid** | 671 | 248 | 682 | 2271 | 3462 | 4035 | 4814 | 4045 | 474 | 767 | 854 | 640 | 3344 | 4082 | 4320 | 4441 |
| **test** | 903 | 664 | 1007 | 2445 | 3831 | 4855 | 5483 | 5029 | 463 | 1090 | 1243 | 438 | 4433 | 5060 | 4866 | 5637 |

Table 12: Data statistics (number of sentence pairs or thousands of sentence pairs for training data) for languages used in multilingual translation experiments.

models. We learn a Sentencepiece vocabulary of size 40k on the training data. We evaluate the models after 100k updates (as opposed to [8] who train for 1.5 million updates) with a per-GPU batch size of 8192 tokens and 32 GPUs. Note we do not use language-aware latent variables despite the multilingual training data. We use the Adam optimizer with learning rate of either 5e-4, 2.5e-4 (24 or 48 layers) or 1.25e-4 (96 layers) and linear warmup followed by polynomial decay with either 5000 (24 or 48 layers) or 15000 (96 layers) warmup steps. For our static model with 96 layers we further tried increasing warmup to 30000 steps and decreasing the learning rate to 1.5625e-5 but this did not help with training loss divergence issues. When using LayerDrop we use 50% dropout probability. We re-use all other hyperparameters from XLM-R [8] (i.e. token masking probability etc.).

**Multilingual Machine Translation.** For multilingual experiments, we use a single Transformer network shared across all languages for both the encoder and decoder, with the same model size as used in [32]. We use the same optimization hyperparameters (learning rate, warm up schedule, etc) as used in WMT English-German experiments except that the batch size is 4096 tokens per-language and we train the model for 14k updates, and evaluated on the last checkpoint. Similarly, we use beam search with beam size 5 and length penalty 1.0 for decoding.

# D   Visualizations

We provide visualizations of the layer selection samples $z_l$ to further illustrate modeling choices around $\mathcal{L}_K$ and priors.

**Effect of $\mathcal{L}_K$.** First, we show that adding the auxiliary loss $\mathcal{L}_K$ discretizes the samples and achieve the pruning purpose by enforcing sparsity of the resulting model. In Figure 8, we visualized samples throughout training using the WMT'16 English-German dataset. Since decoder depth directly contributes to latency at inference time, we only apply $\mathcal{L}_K$ with $K = 12$ to latent layers training in decoder and not in encoder. We could see that samples $z_l$ in decoder becomes discrete throughout training while samples in encoder stay continuous.

Figure 8: Layer selection samples throughout training evaluated on the WMT'16 English-German dataset. Rows correspond to samples from encoder and decoder with 36 latent layers at epoch 2, 6, 25, 50, and 100 respectively. $\mathcal{L}_K$ ($K = 12$) was applied to decoder only and not encoder to contrast the discretizing and pruning effect.

**Effect of priors.** In Section 5 we showed the difference between using an uniform prior Beta(1,1) and aggregated posterior $\tilde{q}(z)$ in the early stage of training. In Figure 9, we further compared the resulting samples used at inference time, where we can see that using aggregated posterior $\tilde{q}(z)$ leads to more consistent sampling behavior for each layer (either "all select" or "all skip") across languages and thus obtain increased sparsity and a more compact model. We used the O2M-Related language group for evaluation, where we could observe qualitatively common layer selection patterns for languages of the same language family, e.g. aze (Azerbaijani) and tur (Turkish), bel (Belorussian) and rus (Russian), glg (Galician) and por (Portuguese), slk (Slovak) and ces (Czech). We leave a systematic study of layer selection and linguistic similarity to future work.

Figure 9: Layer selection samples $z_l$ at inference time trained with uniform prior (left) and aggregated posterior $\tilde{q}(z)$ (right) in $\mathrm{D_{KL}}$. Compared to the uniform prior, using aggregated posterior is more effective for "pruning" by encouraging consistent "select" and "skip" across languages. For example, layer 0, 2, 6, and 23 can be complete pruned for all languages besides language-specific pruning (e.g. for each language/row, layers corresponding to lighter cells could be pruned to derive a sub-network for the given language). This property is appealing for deploying one shared multilingual model for all languages.

## Footnotes

[1]The authors of [31] provided the downloadable data at `https://drive.google.com/uc?export=download&id=0B_bZck-ksdkpM25jRUN2X2UxMm8`

[2]The authors of [32] provided the downloadable data at `https://drive.google.com/file/d/1xNlfgLK55SbNocQh7YpDcFUYymfVNEii/view?usp=sharing`