[Reviews · NeurIPS 2020]

Review 1

Summary and Contributions: In order to train much deeper Transformer-based architectures, the authors propose jointly learning the parameters of the model along with a posterior distribution over the layers of that model, representing which ones to select or drop out for a given task. This distribution is approximated with a Gumbel-Softmax and is used to softly weight layers during training and to prune them at inference time, significantly reducing the runtime / memory usage. The authors show that this proposed method allows them to train deeper models (up to 100 layers) without divergence and to do marginally better on masked language modeling and multilingual machine translation.

Strengths: Novel technique for training deeper modes that do better on MLM and multilingual machine translation tasks. Interesting exploration of multitask machine translation, where a subsequence of layers are used for different language pairs. Motivates layer pruning at inference time with the observation that the runtime at this stage is directly proportional to the number of layers. The auxiliary loss that they added to encourage an effective utilization of k layers seemed clever and effective. Explores the role of latent layers in the decoder, encoder, or in both.

Weaknesses: Doesn't compare against simpler alternatives, such as one described by the authors themselves: instead of learning a separate distribution over layers, pass in the language as an embedding and have the model implicitly learn to weight layers. The authors suggest that this would require additional (Nxd) parameters, but could allow for greater cross-lingual learning. Although the authors are able to train transformer models with up to 100 layers, it's not clear that this is providing any benefit, either in terms of individual task performance or in terms of being a better multitask model (e.g. for the MLM task, LL is the only method that doesn't diverge for the 96 layer model, but its performance is worse than a static 48 layer model; for O2M multilingual translation, LL-D 24/24 outperforms 12/100 in all but en-bul). Very few ablation studies were performed (only one exploring the impact of different loss terms).

Correctness: The claims and methods seem correct and the tasks on which they evaluated their methods (MLM and multilingual MT) are relevant.

Clarity: The paper is well written overall, but there were a number of grammatical errors and typos, including: On page 2, "WMT’16 English-German machine translation task task, masked language modeling ," ['task task', and super nit, extra space before comma] On page 3, "Coupled Variaitonal" "each of which has appealing property to achieve" -> "each of which has the appealing property of achieving" "we let each language to learn" -> "we let each language learn" On page 4, "to demonstrate the effectiveness of the proposed approach at enabling training deeper Transformers [and?] whether this increased depth improves model performance." "WMT'16 Engligh-German sentence pairs" "from averaging [the] last 5 checkpoint[s]" On page 5, "Attenton" is misspelled five times "Next, we compared the learning curves when training deeper models model" (probably don't mean to have the second "model") On page 6, "In Table 1 we evaluating.." (we are evaluating or we evaluated?) "summerized" -> "summarized" On page 7, "an uniform prior Beta" -> "a uniform prior Beta" "In order to understand how the D_KL loss term affect[s] layer selection policies..." "In Table 8 we compare [a] deeper model..." On page 8, "to prevent explode or vanishing gradient" -> "exploding"

Relation to Prior Work: Yes, and the authors compare against prior methods, including DLCL, LayerDrop, and ReZero. However, it would be good to see how deep of a network can be trained with Zhang et al (2019)'s initialization technique and/or how that interacts with the proposed method.

Reproducibility: Yes

Additional Feedback:


Review 2

Summary and Contributions: The paper describes an architecture for building multi-lingual machine translation models in which a transformer network is shared by the language pairs, but different layer depth may be learned by individual language pairs at training time. The paper claims that this approach solves the vanishing gradient problem encountered in training very deep networks. This has the benefit of allowing low resource languages to benefit from deeper network architectures than would otherwise be possible. The main drawback of this paper is the lack of discussion in terms of existing work in multi-lingual machine translation that might permit comparison. [I have read the author rebuttal. Note my comment below on Zhang 2020. ]

Strengths: What is presented in this paper is a respectable idea, one that is probably applicable in other areas (e.g., multi-task learning), namely that large number of transformer layers can be learned without running into the problem of vanishing gradients. The idea that the number of layers can be learned is not itself entirely new; however, the idea that this can be differentially learned for different language pairs in multilingual translation does appear to be novel. The analysis is generally good, and the discussion useful.

Weaknesses: The biggest weakness of this work lies in its lack of comparison with existing machine translation models. This makes it very difficult to assess the relative contribution of this work to existing state of the art systems.

Correctness: Reporting of average bleu scores in Tables 3, 4, 5 and 6 should probably be accompanied by a confidence interval, or some other metric of statistical variation. (Ditto tables 7 & 8) The average scores are for the most part very close, and the differences may not be statistically very significant. Adding or removing a random language pair or two might well change the rankings. In table 1 some bleu scores are annotated with an unspecified measure of variance--it is good that this is done, but what is it? is this a confidence interval? a standard error?

Clarity: The paper is mostly reasonably clear and well written.

Relation to Prior Work: The following paper should probably be addressed in that it offers an different approach to the vanishing gradient problem in deep transformers: Zhang et al. 2019. Improving Deep Transformer with Depth-Scaled Initialization and Merged Attention. EMNLP. I understand that the present paper is couched in terms of the contribution of the algorithm to very deep transformer models. I am surprised, however, at what appears to be little awareness of previous work on multilingual and zero shot machine translation. The authors probably should refer to Zhang et al. 2020. Improving Massively Multilingual Neural Machine Translation and Zero-Shot Translation for the references therein. [As the authors point out correctly in their rebuttal, this paper came out after the submission deadline. It was my intention that they should consult the bibliography of this paper for recent work.]

Reproducibility: Yes

Additional Feedback: The "broader impact statement" doesn't really capture what is intended by "broader impact." A statement of what benefits it brings to machine translation quality, for example, would be more appropriate. All papers listed as being on arXiv need to be checked for acceptance at conferences.


Review 3

Summary and Contributions: The authors present a probabilistic setup to learn which layers in a very deep transformer (64-100 layers) at test-time per-language for multilingual machine translation. They test this method on single language pair translation (WMT16 En-DE), Masked language modeling, and multilingual many to one and one to many translation. The impact of latent weights on gradient scaling is investigated, demonstrating deeper models can be trained without layer normalization.

Strengths: Training deep transformers is a difficult problem, for which multiple methods have been proposed (e.g. layerdrop, rezero). Authors introduce another method, particularly in the case of multilingual translation. Method allows for learning to share or not-share layers across languages. Method allows for targeting a latent depth (compute budget). Empirical evaluations demonstrate consistent significant improvement on multilingual MT, as well as inconsistent improvements on masked language modeling and single-language translation (particularly w.r.t parameter count). Detailed analysis of learned latent layer selection coefficients is conducted.

Weaknesses: Masked Language Modeling: While the method does stabilize training at larger depths, it achieves slightly worse performance. It's unclear whether this is due to dataset size vs. model capacity. An experiment on layer sizes 24, 48, 96 with different dataset sizes might help to illuminate this.

Correctness: Yes. Authors also run several ablations investigating the effect of latent layer selection on (encoder vs decoder vs encoder+decoder), effect of prior, as well as the effects of the various sub-losses

Clarity: Yes, Setup is well explained, and figures in analysis are helpful for understanding.

Relation to Prior Work: Yes, to my knowledge.

Reproducibility: Yes

Additional Feedback: Including some measure of compute time/# of layers selected/# of parameters used in inference in table 1 could help readers understand the performance differences between models. In the current setup, layer selection is dependent only on the language. Could making it dependent on the input as well improve performance? It would be interesting to see a quantitative evaluation of to what extent different languages use the same layers, for both related languages and un-related languages. Would it be possible to compare performance of latent depth multi-lingual models to non-latent depth models of the same depth on single high-resource language pairs? L196: "is also shown effective" -> "is also shown to be effective" The results in the rebuttal on latent depth vs. multi-lingual and hamming distance on layer selection for similar and dissimilar languages further supports the authors' motivation.


Review 4

Summary and Contributions: The paper introduces a probabilistic framework to select which layers to use per language pair in a multilingual machine translation setup. This is done by introducing a binary latent variable z per layer that indicates whether the corresponding layer is used in multilingual NMT. When z is continuous in (0, 1), it acts as gradient scaling that enable training a deep transformer for bilingual NMT. The model is trained end to end via optimizing ELBO. The paper shows that a very deep NMT (up to 100 layers in the decoder) can be trained.

Strengths: - The proposed approach using latent variable z to select Transformer layer or to scale gradient is intuitive and principled. - The experiments are comprehensive, covering both bilingual NMT and multilingual NMT. - Analysis of priors and coefficient parameters is provided to help understand several choices of the model design.

Weaknesses: While increasing the depth of the transformer bring extra gain in BLEU, for bilingual NMT, deeper models also increase the inference time.

Correctness: The claim and method presented in the paper are correct.

Clarity: The paper is well written.

Relation to Prior Work: Yes.

Reproducibility: Yes

Additional Feedback: I wonder what the training time and inference time of the latent depth transformer in comparison to the baselines used in the paper. == Post-rebuttal comment == I keep my score after reading the author response.

[Author Response · NeurIPS 2020]

We thank all reviewers for their thorough reviews and insightful feedback! We are encouraged that they found the proposed idea to be impactful (all reviewers), clear (all reviewers), novel (R1,R2), principled (R3,R4) and applicable to other areas (R2). Specifically, all reviewers recognized our contribution to training deeper models which is an important problem to tackle and the resulting improvement on three sequence modeling tasks over current state-of-the-art (SOTA) approaches. We are also encouraged that the benefit of harvesting a sparse model with reduced inference cost was appreciated (R1,R3). We appreciate that reviewers found our experiments comprehensive and sound (R1,R3,R4), especially acknowledging our comparison to prior work (R1,R3) on training deep models as well as with sufficient ablations and discussion (R2,R3,R4) on modeling choice and sensitivity of hyperparameters. Below we address specific questions which permit more discussions. We will incorporate all suggested improvements in the final version.

**[R2] "The biggest weakness of this work lies in its lack of comparison with existing machine translation models."**
We appreciate you emphasizing the importance of comparing to SOTA models, which we could not agree with more and we'd like to emphasize that we **did** compare to several existing SOTAs. First, we compared the standard Transformer with increased width (Table 1,3,4) which has shown strong performance in bilingual and multilingual MT (Arivazhagan et al. (2019)). Second, we also compared to prior approaches for training deeper ($> 30$ layers) Transformers (e.g. DLCL and LayerDrop in Table 1∼ 5). We did not compare to Zhang et al. (2019) because (1) our method is independent of initialization and thus different angles (just as Zhang et al. (2019) did not compare to non initialization-based methods; (2) most results in Zhang et al. (2019) on deeper models are 12 layers which we found did not diverge and our focus was on $> 24$ layers. We missed Zhang et al. (2020) since it was published at ACL'20 which is *one month after* our submission. But we will include both and relevant multilingual MT references within it in the final version.

**[R2] "unspecified measure of variance" in Table 1.** It is the standard error after running with different seeds.

**[R1] "24/24 outperforms 12/100"** This is a correct statement, however, it results from a combined effect of increased decoder depth and reduced encoder depth. In Table 4, we compared 12/100 (24.16 BLEU) to 12/24 (23.7 BLEU) so as to isolate the effect from increased encoder depths. We did found increased encoder depths brings improvement (especially for low resource) and balanced encoder/decoder depth brings consistent gains as is illustrated in Fig 1. We did not include it in the submission due to the space limit, but we will add it in our final version.

**[R1] "how initialization techniques interacts with the proposed method"**: Our approach solved the same problem (i.e. gradient vanishing) as initialization-based approaches but from a different angle. In addition, it does not require specific initialization (which is usually architecture-dependent) but instead can work with different initializations thanks to the implicit gradient re-scaling effect from latent layers. Also, it enables faster training and inference by learning a shallower network.

Figure 1: Quality improvement (over static depths 12/12) by allocating increased capacity to all-encoder (36/12), all-decoder (12/36), and balanced (24/24).

**[R1] Additional ablation studies.** We had ablation studies on different loss terms as our approach has few bells and whistles. We provided analysis on the effect of different modeling choices and hyperparameters (in Section 5 and the Appendix) which R2, R3 and R4 found useful.

**[R3] "Would it be possible to compare the performance of latent depth multi-lingual models to non-latent depth models of the same depth on single high-resource language pairs?"** Yes. We have done the suggested comparison on the fra-eng (high-resource pair) from the TED corpus where we compared to the strongest bilingual models (12-layer since static depths diverges for deeper models). Our approach outperforms for both directions: fra-eng 40.25 (bilingual static depth 24/12), 40.0 (multilingual static depth 24/12), 41.2 (multilingual latent depth, 24/24); eng-fra 40.22 (bilingual static depth 24/12), 40.3 (multilingual static depth 24/12), 41.5 (multilingual latent depth 24/24).

**[R4] "deeper models also increase the inference time."** This is true for deeper models with static depth. However, our approach allows pruning to a shallower network (especially decoder depth which contributes to the majority of the inference time) and thus addresses the above challenge.

**[R3] "It would be interesting to see a quantitative evaluation of to what extent different languages use the same layers"** We have computed the Hamming distance of layer selections (among 36 layers) between related ($9.33 \pm 1.24$) and unrelated languages ($17 \pm 2.34$) respectively, where we can see the former has a higher degree of parameter sharing.

**Depend on language embeddings [R1] or input [R3]**: Good points! As we elaborated in L104-L108 about the trade-offs, we chose to obtain clear learnings without being confounded by the quality of these additional parameters. But our approach can be easily extended to use both and we fully agree with exploring them in future work.

# References

N. Arivazhagan, A. Bapna, O. Firat, D. Lepikhin, M. Johnson, M. Krikun, M. X. Chen, Y. Cao, G. Foster, C. Cherry, et al. Massively multilingual neural machine translation in the wild: Findings and challenges. *arXiv preprint arXiv:1907.05019*, 2019.

B. Zhang, I. Titov, and R. Sennrich. Improving deep transformer with depth-scaled initialization and merged attention. In *Proceedings of the 2019 Conference on Empirical Methods in Natural Language Processing and the 9th International Joint Conference on Natural Language Processing (EMNLP-IJCNLP)*, pages 897–908, 2019.

B. Zhang, P. Williams, I. Titov, and R. Sennrich. Improving massively multilingual neural machine translation and zero-shot translation. *arXiv preprint arXiv:2004.11867*, 2020.


[Meta-Review · NeurIPS 2020]

The paper introduces a probabilistic approach to learn to select layers of a deep transformer for a language pair in a multi-lingual translation setup. The reviewers found the approach interesting and the potential applications of the technique useful. I would recommend that the paper be accepted. The authors should address the reviewer comments in the final version.